# Spatial distribution and temporal trends in social fragmentation in England, 2001–2011: a national study

Christos Grigoroglou,[1,2] Luke Munford,[2,3] Roger T Webb,[2,4,5] Nav Kapur,[2,5,6,7] Darren M Ashcroft,[2,5,8] Evangelos Kontopantelis[1,2]

For numbered affiliations see end of article.

**Correspondence to**
Mr Christos Grigoroglou;
christos.grigoroglou@
manchester.ac.uk

## ABSTARCT

**Objective** Social fragmentation is commonly examined in epidemiological studies of mental illness as high levels of social fragmentation are often found in areas with high prevalence of mental illness. In this study, we examine spatial and temporal patterns of social fragmentation and its underlying indicators in England over time.

**Setting** Data for social fragmentation and its underlying indicators were analysed over the decennial Censuses (2001–2011) at a small area geographical level (mean of 1500 people). Degrees of social fragmentation and temporal changes were spatially visualised for the whole of England and its 10 administrative regions. Spatial clustering was quantified using Moran's I; changes in correlations over time were quantified using Spearman's ranking correlation.

**Results** Between 2001 and 2011, we observed a strong persistence for social fragmentation nationally (Spearman's r=0.93). At the regional level, modest changes were observed over time, but marked increases were observed for two of the four social fragmentation underlying indicators, namely single people and those in private renting. Results supported our hypothesis of increasing spatial clustering over time. Moderate regional variability was observed in social fragmentation, its underlying indicators and their clustering over time.

**Conclusion** Patterns of social fragmentation and its underlying indicators persisted in England which seem to be driven by the large increases in single people and those in private renting. Policies to improve social cohesion may have an impact on the lives of persons who experience mental illness. The spatial aspect of social fragmentation can inform the targeting of health and social care interventions, particularly in areas with strong social fragmentation clustering.

## Strengths and limitations of this study

► This study uses data over two time points for the whole of England and is the first to describe spatial and temporal patterns of social fragmentation.

► Spatial aspects of social fragmentation can inform organisation of mental health services and social interventions that aim to enhance social support, particularly in areas where social fragmentation is high and is spatially clustered.

► Over the last two decennial Censuses (2001 and 2011), the index of social fragmentation in England strongly persisted at a small area geographical level while its clustering and its underlying indicators increased over time.

► Over time social fragmentation remained relatively stable, but we found marked increases in the percentage of single people and the percentage of those living in private rented accommodation.

► Even though Census data are considered of high quality, people's perception of social fragmentation may not be fully captured in the Census while demographic factors may capture social fragmentation in some areas but not in others.

## INTRODUCTION

The impact of local area characteristics, such as material deprivation, on health is well established.[1][2] However, similar to socioeconomic position, the social networks and the amount of social support in a neighbourhood may also partially determine an individual's health.[3–6] In this context, an area-level characteristic that has been the focus of much investigation in neighbourhood health research is social fragmentation. Social fragmentation measures lack of social cohesion, and it is used to define areas with a breakdown or absence of social capital, and was originally created to measure the non-economic deprivation aspects of areas.[7] Social fragmentation is usually measured using Census data which offer a great potential for the design and implementation of mental health strategies when individual-level data are not available.

Social fragmentation is distinct conceptually from deprivation since fragmentation is a feature of household composition and demographic structure and is not intrinsically linked to socioeconomic position.[8] However, the two measures tend to be positively correlated as they are both concentrated in large towns and cities.[8][9] Thus, many urban localities are both deprived and socially fragmented, although the two phenomena do not always coexist. Several

studies suggest that fragmented communities provide less stable social institutions and social bonds, although maintaining stable institutions and bonds within the communities can contribute substantially to the creation of shared identities, persistence of social relations, promotion of healthy behaviours and good physical and mental health for the population.[10–14]

Social fragmentation has received less attention from researchers than deprivation despite the fact that it can aid the understanding of mechanisms via which the wider social environment influences mental health.[15] Research on adults has shown that social fragmentation is more closely related to mental health than physical health outcomes, independent of material deprivation and individual-level risk factors.[7 16] In the context of mental health, social fragmentation has been linked with suicide and non-fatal self-harm,[7 17–19] mental disorders,[4 11 20] psychiatric health service use,[21] first admission for psychosis,[20–22] psychological distress[3 11] and schizophrenia.[15 23] Nonetheless, physical illness outcomes do not appear to be as clearly linked to social fragmentation.[18 24 25]

In the UK, an index of social fragmentation was developed to measure neighbourhood-level conditions which affect lack of opportunities for social integration.[7] The index is based on indicators of population turnover, percentage of single people, percentage of the population in private renting and percentage of one person households using data from the UK Census. The rationale for the selection of the underlying variables used for the calculation of the index was that several studies, both at ecological and individual level, have found raised mental illness prevalence in highly urbanised areas with high population turnover, many non-family households and reduced social cohesion.[26]

Social fragmentation in a population is likely to have profound effects on its health and well-being, and yet it is incompletely examined and reported in the existing literature, and a number of important research questions remain unanswered. First, how strongly clustered is social fragmentation? Second, how strongly does social fragmentation cluster and change at regional level over two time points? Third, does social fragmentation persist over time? Answering all three questions is important to inform the organisation of healthcare services and to target social and healthcare interventions, especially for mental health for which social fragmentation appears to be a salient risk factor. In this study, we temporally and spatially describe the properties of social fragmentation in England and within each of the 10 administrative English regions and make comparisons across them. Finally, we quantify the persistence of social fragmentation over the study period.

## METHODS
### Data sources
To derive the index of social fragmentation, we accessed decennial Census data from the Nomis website[27] on (1)

single people (including widowed, divorced and separated people), % all usual residents aged ≥ 16; (2) one person households, % of all Census households; (3) households in private renting, % of all Census households and (4) population turnover (percentage of the population that moved into the area from within the UK and from outside the UK), % Census population in the year preceding the census. Our analysis was conducted at a small area geographical level, namely the Lower Layer Super Output Area (LSOA). LSOAs are geographical areas with a mean population of 1500 people. All data were available at the LSOA level except for the population turnover data which were available at the output area level (OA). OAs provide the finest spatial scale at which the Census is enumerated. We assigned the OA-level data to the LSOA level using lookup tables provided by the Office for National Statistics (ONS). More details on the data used to derive each social fragmentation underlying indicator are provided in the online supplementary appendix 1 .

The social fragmentation index for each LSOA was calculated by adding the unweighted z-scores for each of the four characteristics into a composite score in each Census year pooled. We used two time points for the index of social fragmentation, that is, 2001 and 2011 over the two decennial Censuses. Across LSOAs the scores for the index ranged from −6.51 to 26.88 for 2001 and from −7.09 to 25.79 for 2011, with the upper boundaries indicating the most socially fragmented areas and the lower boundaries indicating the most socially cohesive areas. Since the data used to derive the index in 2001 were reported using 2001 LSOAs, we used a weighted means algorithm to assign the data for each underlying indicator to 2011 LSOAs using mid-year population estimates that were obtained from ONS.[28] A detailed description on how we attributed 2001 LSOA data to 2011 LSOAs is provided in the online supplementary appendix 1. We calculated the z-scores of all social fragmentation underlying indicators for each of the two time points of interest. Finally, we obtained data on LSOA rural/urban classification at the 2011 LSOA.[29]

### Analyses
Our outcome of interest was the index of social fragmentation and its four underlying indicators measured separately at the 2001 and 2011 national Censuses. We visualised temporal changes in social fragmentation and its underlying indicators with the use of spatial maps for all of England and each of the 10 English regions/strategic health authorities (North East, North West, Yorkshire & the Humber, East Midlands, West Midlands, East of England, London, South East Coast, South Central and South West), which were the highest level of organisation available in NHS England during the study period. To assess whether levels of spatial autocorrelation for social fragmentation changed over time, we compared the values of Moran's I[30] in the two time points for England and each English region to allow within-England comparisons. The

measure can identify the presence or absence of spatial clusters while accounting for the multidimensional and multidirectional nature of spatial autocorrelation. Under a random spatial pattern, a higher than expected value of Moran's I for social fragmentation would indicate that areas with high levels of social fragmentation are clustered and also that LSOAs with high social fragmentation are bordered with LSOAs with similarly high levels of social fragmentation. We also investigated local indicators for spatial autocorrelation (LISA)[31] to identify local patterns of spatial associations. Furthermore, we quantified the persistency of social fragmentation over the two time points by calculating Spearman's correlation between 2001 and 2011. To visualise and compare temporal changes in the mean social fragmentation levels across the 10 English regions, we plotted population-weighted box plots for the main outcome and its underlying indicators. Finally, we used population-weighted box plots to visualise and compare temporal changes in the social fragmentation levels between the 10 English regions. Analyses were performed with Stata V.14.1 and R V.3.3.1. Due to the size of the data set, effectively the whole of England, statistical significance is irrelevant; observed associations of minimal strength would be statistically significant, and thus we focused on effect sizes wherever possible.

## RESULTS

Region-level characteristics of the index of social fragmentation at both time points are provided in table 1. We found strong correlation for social fragmentation at the two time points (r=0.93), using Spearman's ranking correlation to account for the relative nature of the measure. We plotted spatial maps of social fragmentation for the whole of England and for each region for the two time points analysed. We created maps of social fragmentation from a pooled sample in 2001 and 2011 for the whole of England and London in order to facilitate comparisons between areas over time, and we present those in figures 1 and 2, respectively. Furthermore, we present spatial maps of social fragmentation and its underlying indicators at each time point for every region in online supplementary appendix 2. Temporal changes in social fragmentation for England and for all English regions are presented in figure A in the online supplementary appendix 1. We identified a pattern of stability with modest increases over time for regions in the North, and small decreases in social fragmentation for regions in the South. However, temporal changes of English regions for two of the four social fragmentation underlying indicators were much larger, and they are presented in figure 3.

Large changes in absolute levels of single-people and private renting were found in all regions. More specifically, in 2001 the average number of single persons of the total population was 49.1% while in 2011 the figure was 52.8%, indicating a 7.54% increase. Similarly, in 2001 the average number of people in private renting was 8.5% of all English households while in 2011 this figure increased to 16.2%, indicating a 90.6% increase. The largest increases in numbers of single people were found in the North East, Yorkshire and Humber and West Midlands while London had the highest levels of single people. Moreover, the largest increases for private renting were found in London, Yorkshire and Humber and the South Central while London had again the largest numbers of people living in private renting. For the other two social fragmentation underlying indicators, we found that the average number of one-person households both nationally and regionally remained relatively stable. We also

**Table 1** Temporal changes in social fragmentation metrics across regions, 2001–2011

| | North East | North West | Yorkshire and Humber | East Midlands | West Midlands | East England | London | South East Coast | South Central | South West | England |
|---|---|---|---|---|---|---|---|---|---|---|---|
| Social fragmentation, 2001 | | | | | | | | | | | |
| LSOAs, n | 1656 | 4459 | 3293 | 2732 | 3482 | 3550 | 4765 | 2750 | 2569 | 3226 | 32 482 |
| 25thcentile | −2.32 | −2.34 | −2.31 | −2.59 | −2.59 | −2.50 | −0.46 | −2.37 | −2.51 | −2.09 | −2.26 |
| Median | −0.79 | −0.63 | −0.87 | −1.34 | −1.19 | −1.27 | 2.05 | −1.09 | −1.24 | −0.88 | −0.75 |
| 75thcentile | 0.82 | 1.45 | 0.91 | 0.47 | 0.44 | 0.37 | 4.45 | 0.86 | 0.76 | 0.89 | 1.46 |
| Weighted mean | −0.29 | −0.06 | −0.16 | −0.55 | −0.71 | −0.68 | 2.20 | −0.21 | −0.33 | −0.18 | 0.03 |
| SD | 3.13 | 3.29 | 3.37 | 3.22 | 2.82 | 2.81 | 3.56 | 3.34 | 3.28 | 3.33 | 3.37 |
| Social fragmentation, 2011 | | | | | | | | | | | |
| LSOAs, n | 1657 | 4497 | 3317 | 2774 | 3487 | 3614 | 4835 | 2773 | 2609 | 3281 | 32 844 |
| 25th centile | −2.16 | −2.31 | −2.30 | −2.64 | −2.45 | −2.59 | −0.64 | −2.50 | −2.63 | −2.17 | −2.29 |
| Median | −0.58 | −0.51 | −0.78 | −1.27 | −1.01 | −1.30 | 1.58 | −1.19 | −1.27 | −0.86 | −0.70 |
| 75thcentile | 0.97 | 1.62 | 1.07 | 0.58 | 0.65 | 0.39 | 3.98 | 0.76 | 0.62 | 1.02 | 1.43 |
| Weighted mean | −0.01 | 0.16 | 0.04 | −0.43 | −0.45 | −0.67 | 1.83 | −0.25 | −0.41 | 0.02 | 0.09 |
| SD | 3.26 | 3.50 | 3.50 | 3.24 | 2.94 | 2.75 | 3.28 | 3.24 | 3.28 | 3.28 | 3.32 |

LSOAs, Lower Layer Super Output Areas.

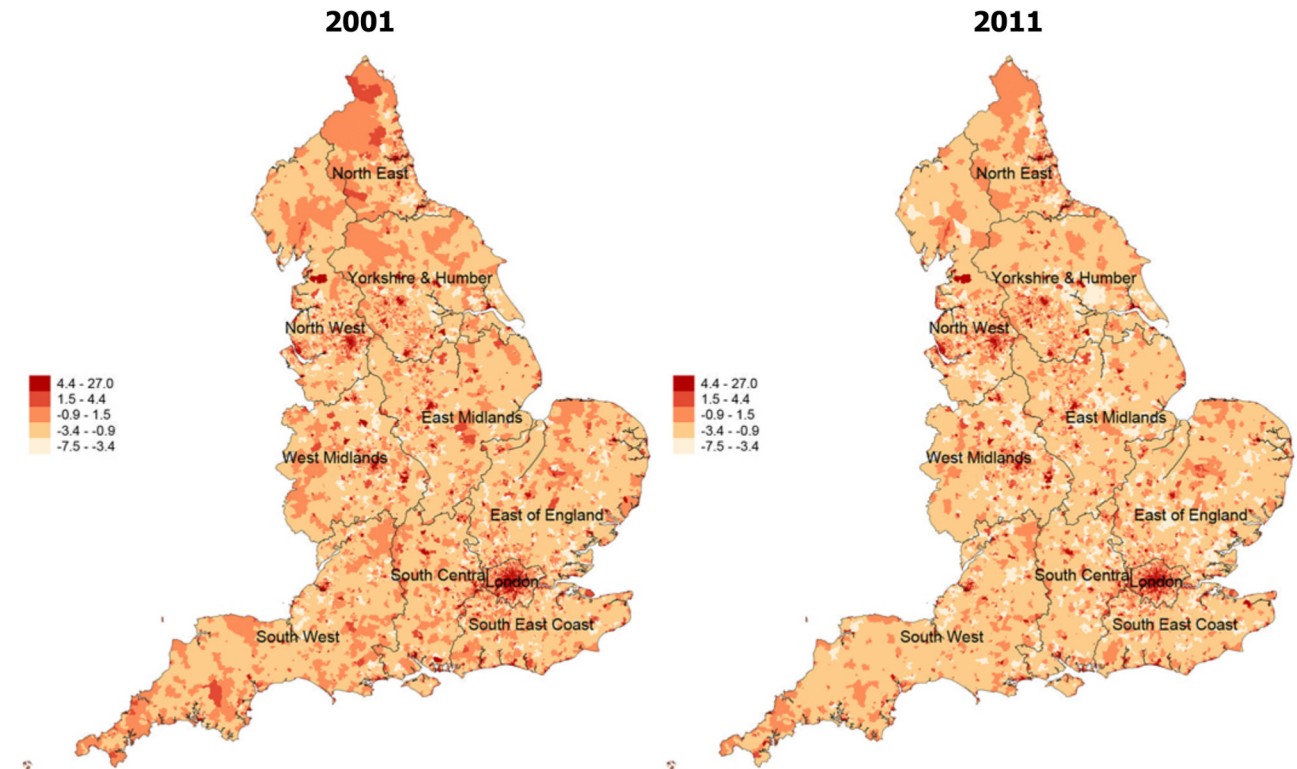

**Figure 1**  Changes in overall social fragmentation for England, 2001 (left) to 2011 (right) (pooled sample).

observed a modest decrease, both nationally and regionally (with the exception of London), in our measure of population turnover which reflects the levels of migration inside and outside areas. Finally, we observed great variability in levels of social fragmentation and its underlying indicators within regions at both time points. High levels of social fragmentation were concentrated in London, the North West and the South West. East England had the lowest levels of social fragmentation followed by West Midlands. There was a clear distinction between rural and urban areas, with the latter having far greater levels of social fragmentation.

Across the whole of England, Moran's I for social fragmentation was very low at 0.0483 in 2001 (95% CI 0.0482 to 0.0485) but by 2011 it had increased to 0.0794 (95% CI 0.0792 to 0.0797), indicating a small increase in spatial autocorrelation and clustering (figure 4). We also observed marked regional variability in social fragmentation spatial autocorrelation levels over time (figure 4). In both time points, the West Midlands had the lowest spatial autocorrelation in social fragmentation, followed by the East of England, while the South West and South East had the highest levels of spatial autocorrelation at both time points. In contrast to increases in spatial autocorrelation across the country between 2001 and 2011, the South East, North East and East Midlands had decreasing levels of spatial autocorrelation over the two time points.

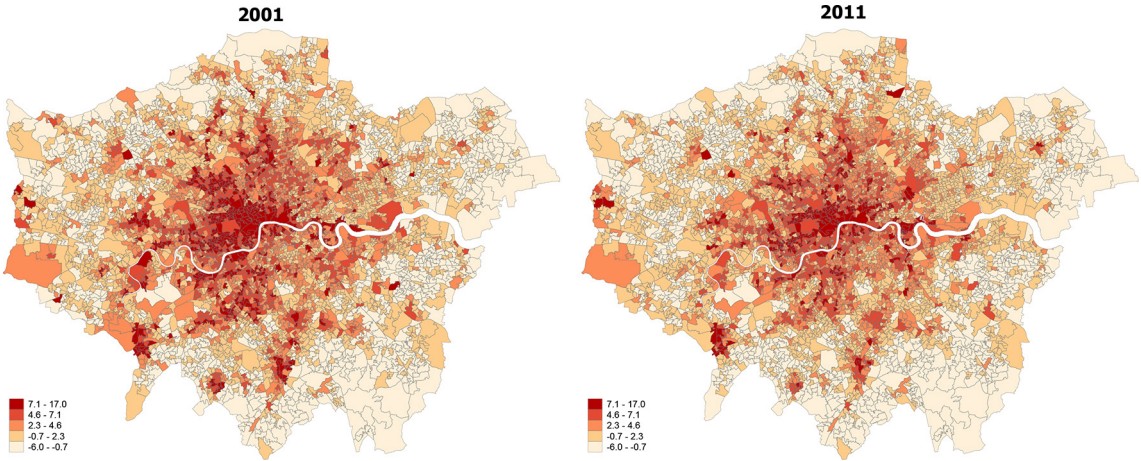

**Figure 2**  Changes in overall social fragmentation for London, 2001 (left) to 2011 (right).

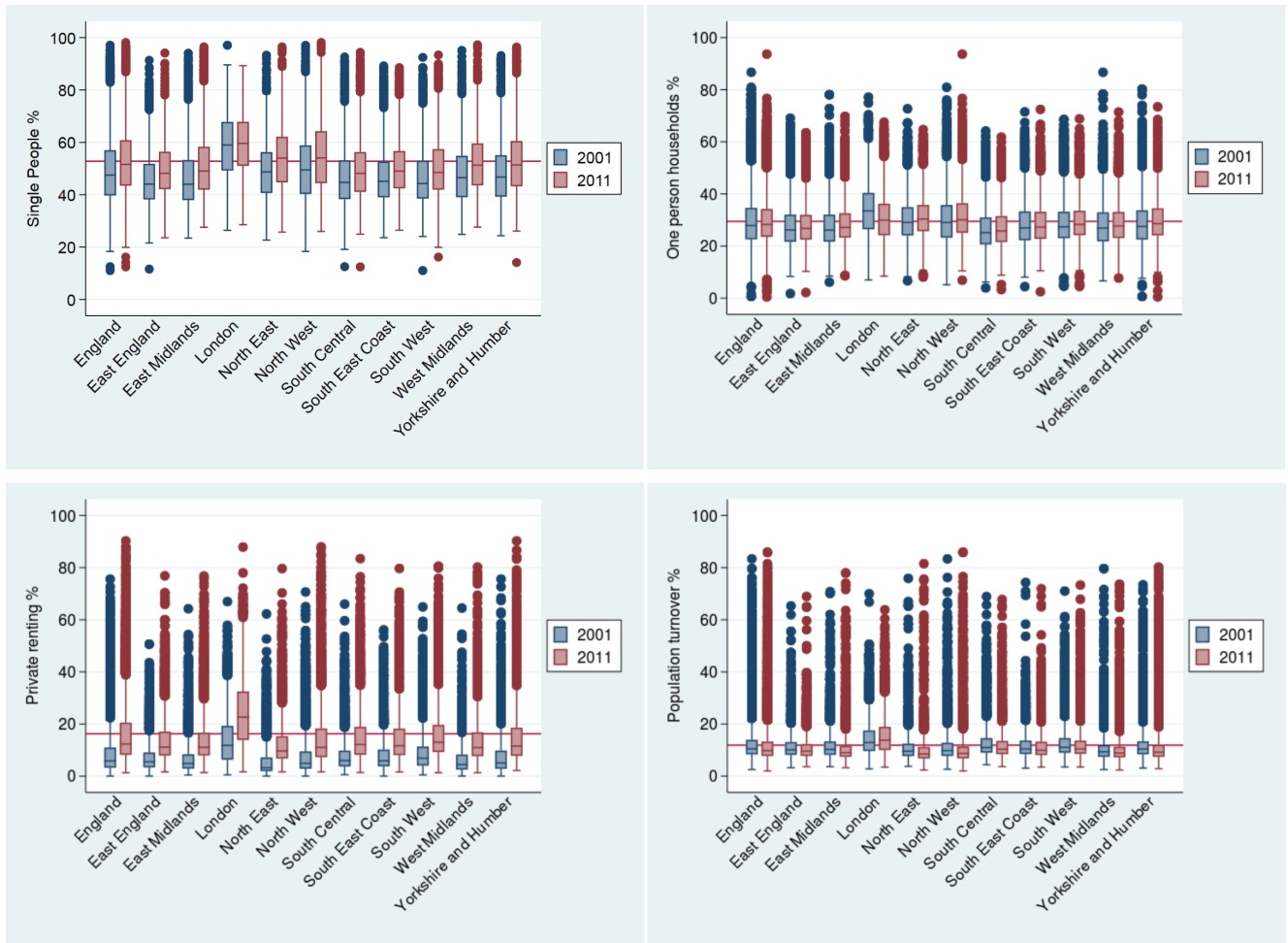

**Figure 3** Percentage of single people, one person households, private renting and population turnover for English regions, over time. *Red line indicates mean for England in 2011.

Spatial clustering also appears to have increased over time for all fragmentation underlying indicators nationally, although its levels remained low (figures A1 and A2 in online supplementary appendix 1). For one-person households, we found very low regional clustering and small regional variation. For single persons and private renting, regional variation in clustering was moderate and their levels were also low with the exception of the South East and South West. Finally, for population turnover we observed the greatest increases of spatial autocorrelation both nationally and regionally but levels of regional clustering were generally low with the exception of South West while regional variability was moderate. We provide graphs and discuss the results from the LISA analysis for local spatial autocorrelation in the online supplementary appendix 1 (figures B1 and B2).

## DISCUSSION

Between 2001 and 2011, we found evidence of strong persistence of social fragmentation in England at small area level. Regional changes over time were modest for social fragmentation, which is not surprising considering the large population denominators and the relative nature of the measure. Similarly, we observed modest increases over time for one-person households and population turnover. However, changes over time for single persons and private renting were much greater, with substantial increases in both underlying indicators across regions, and these specific influences appear to have largely driven the heightened levels of social fragmentation overall. Furthermore, our findings provide evidence of increasing clustering for social fragmentation and its underlying indicators which, in conjunction with the regional variation observed between the two time points, indicates that areas in the South of England (including London) appear much more clustered than the North. Finally, we observed increased clustering for all underlying indicators of social fragmentation over the two time points.

### Strengths and limitations of this study

The main strength of this study lies in its large data set for the whole of England (53 million people in 2011) over

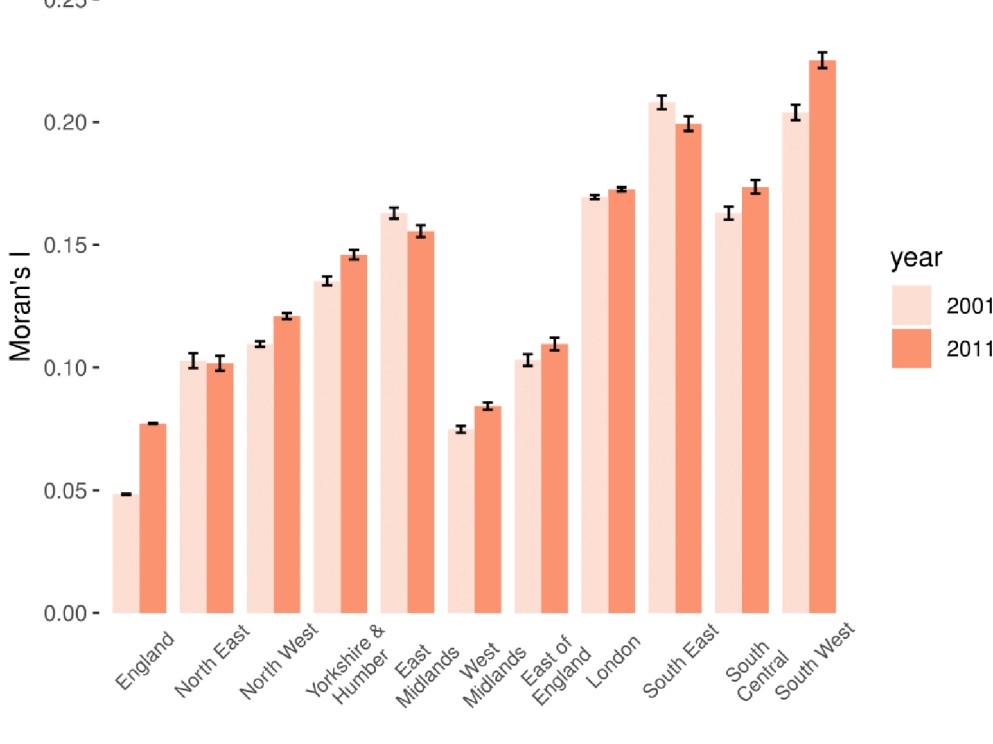

**Figure 4** Spatial autocorrelation with Moran's I for social fragmentation by region, over time.

two time points across 32 844 LSOAs, which we used to investigate spatial and temporal patterns of social fragmentation. To our knowledge, it is the first study of its kind to provide an insight into the temporal persistence of social fragmentation and its clustering.

However, some limitations exist. First, the extent to which the index captures social fragmentation may vary across the country as the demographic factors that we used may not in themselves be valid indicators of social fragmentation or they may adequately measure social fragmentation in some areas but not in others.[22] However, the measure has been reported widely in the literature and on this basis social fragmentation has been identified as an important predictor of mental health outcomes such as suicide. Second, Census questions do not usually directly elicit the presence or quality of neighbourhood social ties or institutions, or people's relationship to their neighbourhood. Therefore, we might have failed to capture people's perception of social fragmentation through the Census data. Nevertheless, we used a measure capturing compositional neighbourhood aspects which appear to be related to the capacity of a neighbourhood to act collectively, and is also important for the construction and maintenance of social ties and institutions. Third, while Census data are recognised to be of a high standard, some unobserved heterogeneity persists.[32] For example, LSOAs are defined from administrative information and this may disguise local effects as contextual characteristics within administrative neighbourhoods are not homogeneous across LSOAs.[33] An alternative approach would require

the use of non-administrative data to define levels, but the Census is currently the only source from which the data on the fragmentation underlying indicators is available. Fourth, for the underlying indicator 'single people' we used data on legal marital status in 2001 and 2011 and registered civil partnership status in 2011. However, the population base for this indicator is not restricted to the household population and this has implications for the total numbers of single people. For example, some of those who are single, widowed or divorced could be living in institutions as well as cohabiting unions which will result in counting cohabitants as 'single, widowed or divorced' in the calculation of the indicator. Fifth, there were boundary changes following the 2011 Census that may have affected our findings,[34] but only 2.5% of LSOAs were affected and we developed an algorithm to make reasonable population weighted-based estimates for these localities, depending on whether they merged, split or some other change occurred. Finally, the index of social fragmentation may serve as a suitable tool to inform allocation of mental health resources in adults, however recent evidence suggests that social fragmentation may not have an effect on children's mental health.[35]

### Interpretation of findings

A previous study showed that between 1971 and 2001 levels of social fragmentation increased steadily with large increases between 1981 and 1991 and modest increases between 1991 and 2001.[36] Our results indicate that this trend was reversed in 2011 as we found a very small

decrease in the overall index of social fragmentation between 2001 and 2011. Furthermore, our results suggest that urbanisation drives the high persistence of social fragmentation in England as we found that levels of social fragmentation and its underlying indicators were consistently higher in urban areas. This may be because urban areas are chosen by diverse social and cultural groups and living in these areas may lead to isolated spaces, lifestyles and problems of social integration. Moreover, we found that the underlying indicators that drive the persistency of social fragmentation appear to be single people and households in private renting, a characteristic of urban environments,[37] as these were the underlying indicators with the largest increases (figure 3).

In absolute terms, private renting was considerably increased in all regions and most notably London; a finding that pertains to unaffordable housing for first-time home buyers and less availability of social housing as a result of the 2008 financial crisis.[38] Similarly, all regions had higher levels of single people in 2011, indicating that numbers of young professionals, students and divorces have increased substantially over the decade. For the other two underlying indicators we did not observe any marked changes across regions, although levels of population turnover appeared to have decreased between the decennial Censuses. The increases we observed in the proportion of single people and those in private renting warrant implications for the prevalence of mental illness especially in urban centres. The private house market is characterised by poor housing conditions as opposed to social houses or owner occupation, and this can have detrimental effects on mental health.[39] Furthermore, it is suggested that married individuals may have better mental health outcomes than single individuals.[40] We believe that these changes are certainly of concern as they may highlight the need for more effective social interventions to target socially fragmented areas in England.

Spatial clustering of neighbourhood social fragmentation appears to have increased over time in England while we observed moderate regional variations in levels of clustering despite the small changes in levels of social fragmentation across regions. Increased levels of spatial clustering for social fragmentation across and within regions can have implications for the planning and organisation of health services as this would imply needs that are not uniformly distributed across a region with the presence of hot spots of high levels of mental illness prevalence, especially in urban areas. To address such spatial health inequalities, services may need to be redesigned locally[41] while infrastructure and spending for mental health needs to be weighted towards social fragmentation hot spots rather than uniformly distributed.

## CONCLUSION

In the absence of individual-level data, area-level indicators are often the only source of information to investigate how factors associated with mental illness may contribute to population health, although when individual-level data are available and are fitted as covariates these area-level associations may be attenuated substantially. Even though social fragmentation is increasingly considered a determinant of mental illness as more epidemiological research emerges that investigates contextual effects on health. To improve mental health in the population, policymakers need to address both relational and structural issues, perhaps by investing in reorganisation of mental healthcare. Solving the problems of cohesion and integration needs to be seen as a collective responsibility across central and local government and all the local agencies. Policies need to address social and educational inequalities by strengthening the economic well-being of areas by lowering local unemployment rates, encouraging populations to bring social support in the communities and encouraging education of the population, which are key strategies for healthy communities. Communities need to work towards tightening social networks and social bonds to help with mental illness prevalence and suicide prevention, for example, by promoting participation in charity, voluntary or community groups and civic engagement. The spatial aspect of social fragmentation is often overlooked, but it can provide vital information for the effective organisation of health services and targeting of health or social interventions. The nature of fragmentation and the mechanisms underlying its association with mental illness are of increasing interest as the population gets older and the healthcare costs associated with mental illness escalate in industrialised countries.

**Author affiliations**

[1]NIHR School for Primary Care Research, Centre for Primary Care, Division of Population Health, Health Services Research and Primary Care, University of Manchester, Manchester, UK
[2]Manchester Academic Health Sciences Centre [MAHSC], Manchester, UK
[3]Centre for Health Economics, Division of Population Health, Health Services Research and Primary Care, University of Manchester, Manchester, UK
[4]Centre for Mental Health and Safety, Division of Psychology and Mental Health, University of Manchester, Manchester, UK
[5]NIHR Greater Manchester Patient Safety Translational Research Centre, Manchester Academic Health Sciences Centre (MAHSC), Manchester, UK
[6]Centre for Suicide Prevention, Division of Psychology and Mental Health, University of Manchester, Manchester, Greater Manchester, UK
[7]Greater Manchester Mental Health Trust, Manchester, UK
[8]Centre for Pharmacoepidemiology and Drug Safety, School of Health Sciences, Faculty of Biology, Medicine and Health, University of Manchester, Manchester Academic Health Sciences Centre (MAHSC), Manchester, UK

**Acknowledgements** The authors thank the Office of National Statistics for the wealth of information they have collected and systematically organised, which made this study possible.

**Contributors** EK, CG, DA and LM designed the study. CG extracted the data from all sources, performed the analyses and drafted the manuscript. EK, RW, DA, LM and NK critically revised the manuscript. CG is the guarantor of this work and, as such, had full access to all the data in the study and takes responsibility for the integrity of the data and the accuracy of the data analysis.

**Funding** This study was funded by the National Institute for Health Research School for Primary Care Research (NIHR SPCR), through CG's PhD. This report is independent research by the National Institute for Health Research. LM acknowledges financial support from the MRC Skills Development Fellowship (MR/N015126/1).

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
