## [Reviewer comments · BMJ Open]

This paper was submitted to a another journal from BMJ but declined for publication following peer review. The authors addressed the reviewers' comments and submitted the revised paper to BMJ Open. The paper was subsequently accepted for publication at BMJ Open.

(This paper received three reviews from its previous journal but only two reviewers agreed to published their review.)

ARTICLE DETAILS

TITLE (PROVISIONAL)	Spatial distribution and temporal trends in social fragmentation in England, 2001 to 2011: a national study
AUTHORS	Grigoroglou, Christos; Munford, Luke; Webb, Roger; Kapur, Navneet; Ashcroft, Darren; Kontopantelis, Evangelos

VERSION 1 – REVIEW

REVIEWER	Peter Congdon Queen Mary University of London, UK
REVIEW RETURNED	28-Aug-2018

GENERAL COMMENTS	This is an interesting paper setting out new findings on the persistence of social fragmentation and extent of spatial clustering. Regarding the latter would there be a benefit to considering/describing the relevance of finer scale analysis using local Moran indices. The variations in SFI by urban status (page 9, line 21) could be made more explicit. For example, what about a Table setting out average SFI scores according to the eight categories of the 2011 Rural and Urban Classification (RUC2011, http://geoportal.statistics.gov.uk/datasets?q=ruc_2011_ Isoa and https://www.gov.uk/government/statistics/2011-rural-urban-classification). Page 5, line 32 (re single marital status), all adult population? Page 6, line 49. subdomain not subdomains Would it be relevant to mention the BBC-University of Sheffield work (http://news.bbc.co.uk/1/hi/uk/7754861.stm) that also looked at changes in SFI.
--

REVIEWER	Heather Joshi Institute of Education University College, Londo UK
REVIEW RETURNED	01-Sep-2018

GENERAL COMMENTS	It was an enjoyable if somewhat frustrating experience to read this paper and its colourful appendix. It was clearly a lot of work to put it together. It is possible that the methods chosen overstate your
--

conclusions. it also possible that they may become more robust after minor clarifications.

P4

I feel that the introductory literature review overstates the case for ecological determinants of health. It does not mention that where individual level information is available the area-level indicators tend to make a minor contribution to explaining differences apparent by individual circumstances. This does not make the census data assembled here uninteresting or irrelevant. Area-level data are often the only source of information, available (or available on a comprehensive basis), and have great potential for designing and implementing policies to prevent and treat mental health problems. It is just the interpretation that needs some more caution. If an ecological correlation is not solely due to the 'ecological' fallacy, it is still worth considering how the causal mechanisms run. Do they go from community to person, or from the factors which determined whether individuals move into or out of the sort of hotspots thrown up by these analysis (or both)? These considerations could also be mentioned in the conclusion.

Page 5 Explain in what sense these data are 'longitudinal' They do not track individuals over time.

They attempt to track LSOAS over two dates. Some attention is paid to the fact that (some) LSOA boundaries change, but I wanted more explanation of how often this occurred and how this was handled in attributing 2001 characteristics to 2011 boundaries. Incidentally, I question the assertion on p 5, 37 that LSOAs are 'administrative'. They are part of Census geography, not genuinely administrative. I suggest you reword the definition.

I would say that indices of social fragmentation are more appropriate for urban areas because they are less well suited to detected the forms of social cohesion and isolation that may obtain in less densely populated areas, not just because they coincide with material deprivation in urban areas..

When you come to introduce your version of the Social Fragmentation Index, it would be useful to give the url of the NOMIS data you used in the references. There is a bit of a minefield in choosing definitions for the elements of this index. (By the way 'sub-domains' seems rather unnecessary jargon, implying some further complexity lying behind each variable). You say 'However, social trends towards increasing cohabitation, a phenomenon that occurs most commonly in urban areas suggest the use of proportion of single, widowed and divorced people instead of married people in the calculation of the index'. The source cited (note 23) does not provide evidence that cohabitation is a mainly urban phenomenon in England, as it refers to a study in Sweden. Perhaps this is a typo? The success of your adaption to avoid cohabitants depends how 'people' and partnership status are defined in the 'single people' indicator. If marital status is 'de jure' rather than de facto the author's index will include cohabitants as 'non-married', which is not the intention. It would be better to cite the census table from which the estimates were taken. It is not clear whether the denominator of this rate, 'people', are those in the household population or the whole population, and what their age range is. If the living arrangements are de facto rather than de jure, the denominator would be the household population, and persons living in institutions would be excluded. If the denominator is de jure marital status, and the population base is not restricted

to those households, some of the 'Single widowed and divorced' could also be living in institutions as well as in cohabiting unions. They might also be children. For the interpretation offered, this variable is probably persons over 16 not living 'couple households', divided by the population aged 16+ living in households, but this is not clearly stated. Line 32 of page 5 'whole population' could be interpreted to include those not living in households and indeed 'people' under 16

The indicator for population turnover needs clearer explanation. Congdon used the proportion of the population who had moved in the last year. The current definition does not specify when the move took place, but suggests it may involve information about crossing OA boundaries. It is not clear whether you count persons or households, or international migrants.

The percentage of households in private rented accommodation links the social fragmentation index to material deprivation. It may have a closer association with economic deprivation in 2011 than it did in the 1990s, given an increase in families with children in this tenure.

It should be stated whether the z scores are calculated by standardising on the standard deviation at any one census, or for each census pooled. Is it done before or after adjustment for boundary change? Similarly are the regional z score time or region specific? The more specific the standardisation, the more will temporal and regional differences be suppressed.

Are regions used in this paper the same as those used by the NHS?

Page 6

Explain what the maps and boxplots show. Give the number of LSOAs plotted (it something like 30,000 rather than the 53m 'population of England'. Explain how (if) Moran's I covers 'clustering' in any sense other than its measuring of spatial clustering, using information on adjacent areas. Is it also affected by the dispersion of each variable in statistical space? In particular I note from scrutinizing the legends to each map that the values allocated to each shade in the heat maps (? Quintiles for England, quartiles for regions) vary across figures. This may exaggerate the impression that little changed over the decade. The account of change over time might be well supplement by a table of changes by LSOAS as well as maps.

Another limit to the success of visualisation is that the maps are not population-weighted. If cartograms were used, the eye would be better drawn to urban areas. As it is the eye is drawn to less densely populated places which occupy a larger surface on the map, and which contain some oddities of scoring high social fragmentation in such places as those with military bases and prisons which should not be automatically assumed to have the same social fabric as zones of transition in inner cities.

Page 7 Did the authors use any particular classification of LSOAs into urban and rural? If so cite.

Pages 8-10

The ensuing discussion may have to be modified in the light of the clarifications I am suggesting.

	The following references may be useful, both have individual as well as ecological data: Vivienne Ivory, Sunny Collings Tony Blakely and Kevin Dew(2011) . When does neighbourhood matter? Multilevel relationships between neighbourhood social fragmentation and mental health. Social Science & Medicine 72(12):1993-2002 'DOI: 10.1016/j.socscimed.2011.04.015 Flouri, E, Midouhas , E, Joshi, H, Sullivan, A. (2015) Neighbourhood social fragmentation and the mental health of children in poverty. Health and Place , pp. 138-145 DOI: 10.1016/j.healthplace.2014.11.009. The latter applies a Congdon-like index to micro and family level data on primary school aged children's behavioural difficulties. It finds very little 'ecological impact' on children's mental health. This is perhaps not surprising since most children live in families, only a few might contribute to the fragmentation index if they live with single parents. It would nevertheless be worth making the point in the discussion (of limitations) that the social fragmentation index would not be a suitable tool for allocating resources to children's mental health. Appendix I suggest the first table in the appendix should be in the main article, if word limits permit. The rest of the appendix ideally needs some introductory commentary, and a warning that it should be used in conjunction with local knowledge. Why the shift from quintiles to quartiles ? Why is only the 2011 map shown for England, while for each region both years are shown? The labelling of the North West region throughout is incorrect. For some maps the colour scheme in densely populated areas breaks down into a grey cloud.
--	---

VERSION 1 – AUTHOR RESPONSE

Reviewer: 1

Reviewer Name: Peter Congdon

Institution and Country: Queen Mary University of London, UK

Please state any competing interests or state 'None declared': None declared

Please leave your comments for the authors below

This is an interesting paper setting out new findings on the persistence of social fragmentation and extent of spatial clustering. Regarding the latter would there be a benefit to considering/describing the relevance of finer scale analysis using local Moran indices.

Response: Thank you for bringing this to our attention. We have now looked at Local Indicators of Spatial Autocorrelation (LISA) to examine spatial autocorrelation at a finer scale. We provide two graphs which depict local Moran scatterplots for social fragmentation at both time points and discuss the results from this analysis in the supplementary material. More particularly we added graphs B1 and B2 and discuss the results from the LISA analysis in p.8-9 of the supplementary material.

1) The variations in SFI by urban status (page 9, line 21) could be made more explicit. For example, what about a Table setting out average SFI scores according to the eight categories of the 2011 Rural and Urban Classification (RUC2011, http://geoportal.statistics.gov.uk/datasets?q=ruc_2011_Isoa and <https://www.gov.uk/government/statistics/2011-rural-urban-classification>).

Response: This is a very good point as we did not look at variations in the index of social fragmentation according to different rural/urban categories in the initial submission. However, the

dataset that you mention is only available for 2011 LSOAs. There is a similar dataset for 2001 LSOAs which includes 6 categories such as 1-2) Town and Fringe 3-4) Village Hamlet and Isolated Dwellings and 5-6) urban areas. However, urban areas are divided in 4 categories in 2011 whereas in 2001 they are divided into 2 categories and a comparison between the two time points may not be feasible due to different classification of urban areas between the two time points. Urban areas in 2001 LSOAs are classified as a) Urban >10k – Less Sparse and b) Urban – 10k – Sparse whereas in 2011 classification of urban areas is more complex as it involves cities, town and small or large conurbations. We could possibly classify areas in both time points in 3 categories (i.e. a) Urban b) Town and Fringe c) Rural Village) but we have not done this as we feel that this classification would not offer much more information than what is currently available with the urban/rural classification.

2) Page 5, line 32 (re single marital status), all adult population?

Response: The denominator in the measure of single people was usual residents aged 16 or over for both 2001 and 2011. We have also revised the manuscript in to explain this. In p.5 par.3 where it read “(2) single people (including widowed, divorced and separated people), % all Census population;” it now reads “(2) single people (including widowed, divorced and separated people), % all usual residents aged 16 or over;”.

3) Page 6, line 49. Sub-domain not subdomains

Response: We have revised the manuscript to make this change.

4) Would it be relevant to mention the BBC-University of Sheffield work (<http://news.bbc.co.uk/1/hi/uk/7754861.stm>) that also looked at changes in SFI.

Response: Thank you. This is indeed an important study that was omitted. We revised the manuscript to acknowledge the findings from this work. In p.9 par.2 we added the following statement “A previous study showed that between 1971 and 2001, levels of social fragmentation increased steadily with large increases between 1981 and 1991 and modest increases between 1991 and 2001.¹ Our results indicate that this trend was reversed in 2011 as we found a small decrease in the overall index of social fragmentation between 2001 and 2011.”

Reviewer: 2

Reviewer Name: Heather Joshi

Institution and Country: Institute of Education, University College, London, UK

Please state any competing interests or state ‘None declared’: none

Please leave your comments for the authors below

It was an enjoyable if somewhat frustrating experience to read this paper and its colourful appendix. It was clearly a lot of work to put it together. It is possible that the methods chosen overstate your conclusions. It is also possible that they may become more robust after minor clarifications.

P4

1) I feel that the introductory literature review overstates the case for ecological determinants of health. It does not mention that where individual level information is available the area-level indicators tend to make a minor contribution to explaining differences apparent by individual circumstances. This does not make the census data assembled here uninteresting or irrelevant. Area-level data are often the only source of information, available (or available on a comprehensive basis), and have great potential for designing and implementing policies to prevent and treat mental health problems. It is just the interpretation that needs some more caution. If an ecological

correlation is not solely due to the 'ecological' fallacy, it is still worth considering how the causal mechanisms run. Do they go from community to person, or from the factors which determined whether individuals move into or out of the sort of hotspots thrown up by these analyses (or both)? These considerations could also be mentioned in the conclusion.

Response: We agree with your point that when individual information is available the area-level indicators make only a minor contribution to explaining differences for individuals. We revised the manuscript and we included the following statement in the Conclusion subsection. "In the absence of individual-level data, area-level indicators are often the only source of information to investigate how factors associated with mental illness may contribute to population health, albeit that when individual-level data are available and fitted as covariates these area-level associations may be attenuated substantially." We hope this clearly illustrates the issue with ecological fallacy when individual-level data are unavailable.

2) Page 5. Explain in what sense these data are 'longitudinal'. They do not track individuals over time.

They attempt to track LSOAs over two dates. Some attention is paid to the fact that (some) LSOA boundaries change, but I wanted more explanation of how often this occurred and how this was handled in attributing 2001 characteristics to 2011 boundaries. Incidentally, I question the assertion on p 5, 37 that LSOAs are 'administrative'. They are part of Census geography, not genuinely administrative. I suggest you reword the definition.

Response: For your first point we agree that the term "longitudinal" may overstate the nature of the dataset. We have now revised the manuscript to replace the terms "longitudinal" and "longitudinally" with the terms "over two time points" and "temporally" respectively, throughout the manuscript.

As far as we are aware, the LSOA boundaries have only been altered once, in 2011, following the 2003 National Statistics policy to minimise the statistical impact of frequent electoral ward boundary changes. In 2011 there was a modification of OAs (Output Areas) and SOAs (Super Output Areas) in England & Wales, mainly due to significant population changes between the decennial censuses. We are not aware of any other more recent change that occurred in OA or SOA boundaries.

In order to attribute 2001 LSOA information to 2011 LSOAs we used population weighted regression modelling to estimate 2001 data at the 2011 LSOA. More specifically, if an LSOA remained unchanged in 2011 (and that was the case for 97.4% of all LSOAs) we took no action and attributed the social fragmentation values from 2001 to the 2011 LSOAs. If two or more 2001 LSOAs merged into one LSOA in 2011 (0.6% of all LSOAs), we calculated the mean value of social fragmentation weighted for 2003 LSOA population (which is the oldest population estimates dataset provided by ONS). If a 2001 LSOA split into two or more 2011 LSOAs (1.8% of all LSOAs) we assigned the relevant 2001 social fragmentation scores. Finally, there were some LSOAs for which changes did not match any of the previously described patterns (i.e. an LSOA split into two and each part merged with a different LSOA). For this very small number of LSOAs (0.1% of all LSOAs) we developed an algorithm to calculate population weighted mean estimates. The corresponding algorithm is available from the authors.

We decided to include the paragraph above in the supplementary material under the title "Spatial weighted regressions to attribute 2001 data to 2011 LSOAs" to provide any interested readers with more details on our methodology. Regarding your point about describing LSOAs as administrative units, we replaced the term "administrative" with the term "geographic areas".

3) I would say that indices of social fragmentation are more appropriate for urban areas because they are less well suited to detect the forms of social cohesion and isolation that may obtain in less densely populated areas, not just because they coincide with material deprivation in urban areas. When you come to introduce your version of the Social Fragmentation Index, it would be useful to give the url of

the NOMIS data you used in the references. There is a bit of a minefield in choosing definitions for the elements of this index. (By the way 'sub-domains' seems rather unnecessary jargon, implying some further complexity lying behind each variable).

Response: We revised the manuscript to provide the url to the Nomis Official Labor Market Statistics in the corresponding reference (i.e. reference 26). We understand that we did not explain the population turnover indicator in much detail in the originally submitted manuscript, and therefore we have now provided additional information on the population turnover and the other three indicators in the Supplementary appendix. We hope this provides clarity in relation to the reviewer's query regarding population turnover and other social fragmentation indices. Following your recommendations we also replaced the term "subdomain" with the term "underlying indicator". We feel that this describes the components of the index of social fragmentation better.

4) You say 'However, social trends towards increasing cohabitation, a phenomenon that occurs most commonly in urban areas suggest the use of proportion of single, widowed and divorced people instead of married people in the calculation of the index'. The source cited (note 23) does not provide evidence that cohabitation is a mainly urban phenomenon in England, as it refers to a study in Sweden. Perhaps this is a typo? The success of your adaption to avoid cohabitants depends how 'people' and partnership status are defined in the 'single people' indicator. If marital status is 'de jure' rather than de facto the author's index will include cohabitants as 'non-married', which is not the intention. It would be better to cite the census table from which the estimates were taken. It is not clear whether the denominator of this rate, 'people', is those in the household population or the whole population, and what their age range is. If the living arrangements are de facto rather than de jure, the denominator would be the household population, and persons living in institutions would be excluded. If the denominator is de jure marital status, and the population base is not restricted to those households, some of the 'Single widowed and divorced' could also be living in institutions as well as in cohabiting unions. They might also be children. For the interpretation offered, this variable is probably persons over 16 not living 'couple households', divided by the population aged 16+ living in households, but this is not clearly stated. Line 32 of page 5 'whole population' could be interpreted to include those not living in households and indeed 'people' under 16.

Response: This was not a typo as we chose to cite the original paper where cohabitation was first mentioned as an increasing phenomenon in urban areas. We agree though, this reference may not be considered relevant for England thus we provide additional information from the Office for National Statistics statistical bulletin which describes the increasing trends in cohabitation among families in England.² Furthermore, several studies³⁻⁵ based in England use the proportion of single or unmarried people in an area instead of the proportion of married people and in our study we followed the same pathway in order to align our study with other studies in the field.

For the definition of single people we used the 'KS004-Marital status' and the KS103UK – Marital and civil partnership status' census tables for 2001 and 2011 respectively. Both census tables use legal marital status and the 2011 table uses also registered civil partnership status therefore marital status is defined as de jure. Thank you for bringing this to our attention. It was a limitation that was overlooked in the original submission. We added the following statement about this limitation in p.9 par. 1 "Fourth, for the underlying indicator 'single people' we used data on legal marital status in 2001 and 2011 and registered civil partnership status in 2011. This has implications for the total numbers of single people, as those who cohabit will be counted as 'non-married' in the calculation of the indicator."

Regarding the denominator of single people, we calculated the percentage of people, not households, who lived alone at both time points and we divided those values by the number of people aged 16 or older in each LSOA at both time points. We agree that the census tables should be cited, and so we have provided all relevant information on the data used and their respective census tables in the supplementary material.

5) The indicator for population turnover needs clearer explanation. Congdon used the proportion of the population who had moved in the last year. The current definition does not specify when the move took place, but suggests it may involve information about crossing OA boundaries. It is not clear whether you count persons or households, or international migrants.

Response: We understand that this was not clear in the originally submitted manuscript and therefore we revised the text to further explain the population turnover indicator. In p.4. par.3 where it read "(3) Population turnover (percentage of the population that moved from another country or within the country), % Census population" it now reads "(3) Population turnover (percentage of the population that moved into the area from within the UK and from outside the UK), % Census population in the year preceding the census".

6) The percentage of households in private rented accommodation links the social fragmentation index to material deprivation. It may have a closer association with economic deprivation in 2011 than it did in the 1990s, given an increase in families with children in this tenure.

Response: The indicator may be more closely linked to socioeconomic deprivation. However, we do not analyse deprivation in this work and the first comprehensive measure of deprivation was the Index of Multiple Deprivation 2004, a few years after the 2001 census. Therefore, we feel that we cannot compare the association between private renting and deprivation in 2001 versus 2011.

7) It should be stated whether the z scores are calculated by standardising on the standard deviation at any one census, or for each census pooled. Is it done before or after adjustment for boundary change? Similarly are the regional z scores time or region specific? The more specific the standardisation, the more will temporal and regional differences be suppressed.

Response: The z-scores are calculated by standardising on the standard deviation for each census pooled. In our study we calculated the proportions of each variable within an LSOA in 2001 and then we assigned those proportions to 2011 LSOAs using an algorithm developed for this purpose. Subsequently we calculated the z-scores for each social fragmentation underlying indicator at 2011 LSOA and then created the composite index. We acknowledge that this was not sufficiently clear in the originally submitted manuscript and in p.5 par.3, where it read "The social fragmentation index for each LSOA was calculated by adding the unweighted z scores for each of the four characteristics into a composite score", it now reads "The social fragmentation index for each LSOA was calculated by adding the unweighted z scores for each of the four characteristics into a composite score in each census year pooled."

We also revised the following I statement to explain this. In the manuscript (p.6 par.1) where it read "we used a weighted means algorithm to assign the data to 2011 LSOAs using mid-year population estimates that were obtained from ONS.²" it now reads "we used a weighted means algorithm to assign the data for each underlying indicator to 2011 LSOAs using mid-year population estimates that were obtained from ONS.² We calculated the z-scores of all social fragmentation underlying indicators for each of the two time points of interest."

8) Are regions used in this paper the same as those used by the NHS?

Response: The regions used in the paper refer to Strategic Health Authorities (SHA) which were the highest geographical organisation level available for NHS England during the study period. However, we feel that this was not sufficiently clear in the previously submitted manuscript and we have therefore revised the manuscript in p.6 par.2. Where it read "10 English regions (North East, North West, Yorkshire & the Humber, East Midlands, West Midlands, East of England, London, South East Coast, South Central and South West)" It now reads "10 English regions/strategic health authorities (SHA) (North East, North West, Yorkshire & the Humber, East Midlands, West Midlands, East of England,

London, South East Coast, South Central and South West) which were the highest level of organisation in NHS England during the study period.”

Page 6

9) Explain what the maps and boxplots show. Give the number of LSOAs plotted (it something like 30,000 rather than the 53m ‘population of England’. Explain how (if) Moran’s I covers ‘clustering’ in any sense other than its measuring of spatial clustering, using information on adjacent areas. Is it also affected by the dispersion of each variable in statistical space? In particular I note from scrutinizing the legends to each map that the values allocated to each shade in the heat maps (? Quintiles for England, quartiles for regions) vary across figures. This may exaggerate the impression that little changed over the decade. The account of change over time might be well supplement by a table of changes by LSOAS as well as maps.

Response: In p.8 par.3, where it originally read “The main strength of this study lies in its large dataset for the whole of England (53 million people in 2011) over two time points,” it now reads “The main strength of this study lies in its large dataset for the whole of England (53 million people in 2011) over two time points across 32,844 LSOAs,”. Moran’s I is by definition a measure of spatial clustering and we did not use it to measure clustering in any other sense. All the aspects of spatial clustering were measured using global and local indicators (which we added after a suggestion by reviewer 1) of spatial autocorrelation (i.e. global and local Moran’s I) whereas the temporal trends were measured using correlation coefficients over the two time points and with the use of spatial maps. The reviewer is right that Moran’s I can be sensitive to outliers and highly skewed distributions; however in our data there was only one outlier, which was the same LSOA at both time points. When we removed that particular LSOA from the datasets at the two time points the values of Moran’s I did not change much. Moran’s I original value was 0.0772 and after we excluded the outlier from the both datasets (i.e. 2001 and 2011 dataset), Moran’s I became 0.0880 in 2001 and 0.0790 in 2011. Therefore, we did not observe any substantial change in the values of Moran’s I due to the presence of outliers. Regarding the values allocated to each shade in the spatial maps, the values represent the levels of social fragmentation in each time point for each region. According to your helpful suggestions we have now also provided a region-level table (Table 1) with the characteristics of the measure across time (median LSOA, 25th and 75th centile LSOA, weighted mean and SD).

10) Another limit to the success of visualisation is that the maps are not population-weighted. If cartograms were used, the eye would be better drawn to urban areas. As it is the eye is drawn to less densely populated places which occupy a larger surface on the map, and which contain some oddities of scoring high social fragmentation in such places as those with military bases and prisons which should not be automatically assumed to have the same social fabric as zones of transition in inner cities.

Response: Thank you for the suggestion. However, LSOAs are constructed in a way that they hold around 1500 people. They changed after the 2011 census, primarily to meet that target. Some variability exists of course, but we feel not enough to be conveyed through a cartogram. However, we have now added a region-level table (Table 1) with the characteristics of the measure across time (median LSOA, 25th and 75th centile LSOA, weighted mean and SD).

11) Page 7 Did the authors use any particular classification of LSOAs into urban and rural? If so cite.

Response: Yes this was overlooked in the original submission. We revised the manuscript and provide a statement in relation to rural/urban classification. In p.6 par.1 we added the following “Finally, we obtained data on LSOA rural/urban classification at the 2011 LSOA.”⁶

Pages 8-10

12) The ensuing discussion may have to be modified in the light of the clarifications I am suggesting.

The following references may be useful, both have individual as well as ecological data:

Vivienne Ivory, Sunny Collings Tony Blakely and Kevin Dew(2011) . When does neighbourhood matter? Multilevel relationships between neighbourhood social fragmentation and mental health. *Social Science & Medicine* 72(12):1993-2002 'DOI: 10.1016/j.socscimed.2011.04.015

Flouri, E, Midouhas , E, Joshi, H, Sullivan, A. (2015) Neighbourhood social fragmentation and the mental health of children in poverty. *Health and Place*, pp. 138-145 DOI: 10.1016/j.healthplace.2014.11.009.

The latter applies a Congdon-like index to micro and family level data on primary school aged children's behavioural difficulties. It finds very little 'ecological impact' on children's mental health. This is perhaps not surprising since most children live in families, only a few might contribute to the fragmentation index if they live with single parents. It would nevertheless be worth making the point in the discussion (of limitations) that the social fragmentation index would not be a suitable tool for allocating resources to children's mental health.

Response: Thank you for highlighting these two additional references. We feel that the work by Ivory et. al is important and we now cite the paper in the second paragraph of the introduction. Furthermore, we have added a statement in the limitations sections to acknowledge that social fragmentation can only be used in allocating resources for adults and not children. In p.9 par.1 we have added the following statement "Finally, the index of social fragmentation may serve as a suitable tool to inform allocation of mental health resources in adults, although recent evidence suggests that social fragmentation may not have an effect on children's mental health."

Appendix

13) I suggest the first table in the appendix should be in the main article, if word limits permit. The rest of the appendix ideally needs some introductory commentary, and a warning that it should be used in conjunction with local knowledge. Why the shift from quintiles to quartiles? Why is only the 2011 map shown for England, while for each region both years are shown? The labelling of the North West region throughout is incorrect. For some maps the colour scheme in densely populated areas breaks down into a grey cloud.

Response: We also think that the first table in the appendix should be in main article; however the journal does not allow more figures than what we currently have in the main article and the addition of Table 1 in the main text may not permit the inclusion of figure A (supplementary appendix). We have added some more material to the appendix as per your suggestions. There was no specific reason for the shift from quintiles to quartiles - this was merely a programming glitch. We have corrected Figure 1 which now only features quartiles. All of the maps included in the manuscript and its supplementary materials now depict quartiles. We have also added the maps for social fragmentation and the underlying indicators for 2001 in the supplementary file 2 and corrected the label for North West throughout. Finally, the "grey cloud" depicts the LSOA borders and because urban areas are so densely populated the borders are more emphasized. This is the thinnest border with which the maps can be presented. We trust that this has no real implications; on the contrary it helps readers distinguish specific areas within densely populated/urban larger areas.

1. Dorling D, Vickers D, Thomas B, Pritchard J, Ballas D. Changing UK. *The way we live now*. Sheffield: University of Sheffield 2008.

2. Office for National Statistics. Super Output Area mid-year population estimates for England and Wales, Mid-2011. 2016.

3. Fagg J, Curtis S, Stansfeld SA, Cattell V, Tupuola A-M, Arephin M. Area social fragmentation, social support for individuals and psychosocial health in young adults: Evidence from a national survey in England. *Soc Sci Med* 2008;66(2):242-54.
4. Allardyce J, Gilmour H, Atkinson J, Rapson T, Bishop J, McCreadie R. Social fragmentation, deprivation and urbanicity: relation to first-admission rates for psychoses. *Br J Psychiatry* 2005;187(5):401-06.
5. Congdon P. Assessing the impact of socioeconomic variables on small area variations in suicide outcomes in England. *Int J Environ Res Public Health* 2012;10(1):158-77.
6. Bibby P, Shepherd J. Developing a new classification of urban and rural areas for policy purposes—the methodology. *London: DEFRA* 2004.
7. Flouri E, Midouhas E, Joshi H, Sullivan A. Neighbourhood social fragmentation and the mental health of children in poverty. *Health Place* 2015;31:138-45.

VERSION 2 – REVIEW

REVIEWER	Peter Congdon QMUL, UK
REVIEW RETURNED	31-Oct-2018

GENERAL COMMENTS	The revision meets the points raised at review
--

REVIEWER	Heather Joshi UCL Institute of Education
REVIEW RETURNED	28-Oct-2018

GENERAL COMMENTS	My fascination with this paper is undimmed, but I am still frustrated by some of the exposition. Please explain to the reader why Figures 1 and 2 show a time constant classification of colours for the heat map. -Are they quintiles on a pooled sample? whereas different cutoffs appear for each year and each region in the maps in the Appendix. The maps for England and London have two sets of cut-offs in the Appendix and one set in Figures 1 and 2. I am wondering if the answer is connected to the question I put in my original review about whether the construction of Z scores used a temporally pooled set of data or was time and region specific. I still think that readers should be told this information somewhere. You have ignored my comments about your original text on cohabitation, on page 5, As you state on page 9, line 29, your approach does not recognize that some de jure 'single' people' are not likely to be socially isolated if they are living as cohabitants. You have also ignored my comments about the awkward case of 'single' people living in institutions. This could be fixed with a little redrafting. YOU could cut the reference to Sweden. I see that you have included some reference to the context/composition issue in the conclusions. I would recommend you elaborate on this theme a little in the introduction. On page 3 of the supplement, please add that migrants are defined as having moved in within the year before the census. On page 9 of the supplement spell out what LISA stands for
--

VERSION 2 – AUTHOR RESPONSE

Reviewer: 1

Reviewer Name: Peter Congdon

Institution and Country: QMUL, UK

Please state any competing interests or state 'None declared': None

Please leave your comments for the authors below
The revision meets the points raised at review

Reviewer: 2

Reviewer Name: Heather Joshi

Institution and Country: UCL Institute of Education

Please state any competing interests or state 'None declared': none declared

Please leave your comments for the authors below

My fascination with this paper is undimmed, but I am still frustrated by some of the exposition.

1) Please explain to the reader why Figures 1 and 2 show a time constant classification of colours for the heat map. -Are they quintiles on a pooled sample? Whereas different cut-offs appear for each year and each region in the maps in the Appendix. The maps for England and London have two sets of cut-offs in the Appendix and one set in Figures 1 and 2. I am wondering if the answer is connected to the question I put in my original review about whether the construction of Z scores used a temporally pooled set of data or was time and region specific. I still think that readers should be told this information somewhere.

Response: We apologise for the confusion with the graphs of social fragmentation. Figures 1 and 2 have the same keys in their legends for both time points, to allow readers to easily compare areas over time. In figures 1 and 2 we divided the maps for 2001 and 2011 into equal quintiles from one pooled sample across the two time points and the reason for doing this is because we have different levels of social fragmentation in 2001 and 2011 and we wanted to see whether areas rank differently in levels of social fragmentation over time. In contrast the supplementary material depicts social fragmentation quintiles from the two distinct samples (one for 2001 and one for 2011), using different cut-off points that are a better suit for levels of social fragmentation at each time point. We understand that this was confusing so we attempted to clarify this in the manuscript. Where it read: "Maps of social fragmentation in 2001 and 2011 and their respective changes for the whole of England and London are presented in Figure 1 and Figure 2 respectively, whilst social fragmentation and its underlying indicators maps over time for each region are presented in supplementary Appendix 2.", it now reads: "We created maps of social fragmentation from a pooled sample in 2001 and 2011 for the whole of England and London in order to facilitate comparisons between areas over time and we present those in Figure 1 and Figure 2 respectively. Furthermore, we present maps of social fragmentation and its underlying indicators at each time point for every region in supplementary Appendix 2." We hope this clarifies the reviewer's question.

2) You have ignored my comments about your original text on cohabitation, on page 5, As you state on page 9, line 29, your approach does not recognize that some de jure 'single' people are not likely to be socially isolated if they are living as cohabitants.

Response: Thank you for your suggestion, however, we disagree that we ignored your comments. You made a very valid point which was taken into account in the first revision but perhaps we should have made our response clearer. We have now expanded a bit more in p.9, line 19 and where it read "Fourth, for the underlying indicator 'single people' we used data on legal marital status in 2001 and 2011 and registered civil partnership status in 2011. This has implications for the total numbers of single people, as those who cohabit will be counted as 'non-married' in the calculation of the indicator." it now reads "Fourth, for the underlying indicator 'single people' we used data on legal marital status in 2001 and 2011 and registered civil partnership status in 2011. However, the population base for this indicator is not restricted to those households for which legal marital status information is available and this has implications for the total numbers of single people. For example, some of those who are single,

widowed or divorcees could be living in institutions as well as cohabiting unions which will result in counting cohabitants as 'single, widowed or divorced' in the calculation of the indicator." Our strategy on dealing with this was to firstly acknowledge that including single, widowed or divorced people may be the best way of measuring single people as it is suggested in the literature and then acknowledge that there are limitations with this measure which you very kindly highlighted.

3) You have also ignored my comments about the awkward case of 'single' people living in institutions. This could be fixed with a little redrafting. You could cut the reference to Sweden.

Response: We have now revised the manuscript to include a statement that refers to this point. In p.9, line 19 it now reads "Fourth, for the underlying indicator 'single people' we used data on legal marital status in 2001 and 2011 and registered civil partnership status in 2011. However, the population base for this indicator is not restricted to those households for which legal marital status information is available and this has implications for the total numbers of single people. For example, some of those who are single, widowed or divorced could also be living in institutions as well as in cohabiting unions which would result in counting cohabitants as 'single, widowed or divorced' in the calculation of the indicator." We have also removed the reference to Sweden as you suggested.

4) I see that you have included some reference to the context/ composition issue in the conclusions. I would recommend you elaborate on this theme a little in the introduction.

Response: Thank you. We have now added the following statement in the introduction (p.4. par.1) "Social fragmentation is usually measured using census data which offer a great potential for the design and implementation of mental health strategies when individual level data are not available."

5) On page 3 of the supplement, please add that migrants are defined as having moved in within the year before the census.

Response: Thank you for bringing this to our attention. We have added that migrants are defined as those who have moved in the area within the year before the census.

6) On page 9 of the supplement spell out what LISA stands for

Response: We have now spelled out the LISA acronym which stands for "Local Indicator Spatial Association" analysis.

VERSION 3 – REVIEW

REVIEWER	Heather Joshi UCL institute of Education
REVIEW RETURNED	12-Nov-2018
GENERAL COMMENTS	Your latest revisions satisfy most of my recent comments, except: On page 5 the first complete sentence should be removed or reworded. You have recognised my point, on page 9 below that the 'Single Widowed and Divorced' include people who are cohabiting. Therefore taking this group instead of the married does not make any allowance for unmarried couples. You will have to decide where to make the references, or delete and renumber.

	Page 9 line 21: for 'those household which marital status is availble' should read 'the household population'. Marital status is actually known for everyone, not just those living in households.
--	--

VERSION 3 – AUTHOR RESPONSE

We wish to thank again the reviewers for their input. It's been a somewhat pleasant experience.

We have now revised the manuscript to take into account the second reviewer's final comments.

More specifically, we deleted the sentence about cohabitation on page 5 and re-numbered the references accordingly while we rephrased line 21 on page 9 following the suggestions.